# On Some Theoretical Aspects of The Evaporation Process of a Droplet and Its Optimal Size When Extinguishing Fires

## Sergey Oktyabrinovich Gladkov

Moscow Aviation Institute, National Research University, Volokolamskoye sh., 4, 125993 Moscow, Russia; sglad51@mail.ru

**Abstract:** We are proposing a model mathematical description of droplet evaporation using the kinetic approach. We have obtained the basic equation of the theory by using the law of conserving the full power of the vapor–liquid system, which has not been done before. We have found the range of droplet sizes at which it is stable. We have given a comparison of the obtained results with the known traditional ones. We have given numerical estimates for the critical size of the fine-dispersed phase up to the value of which ordinary evaporation takes place (that is for Knudsen number $Kn = \frac{l}{R}$, inequality $Kn \ll 1$ must be fulfilled, where $l-$ is the free path of the molecule and $R-$ is the droplet radius). We have given the optimal droplet size which is the most effective from the point of view of technical use in extinguishing flammable oil transformers.

**Keywords:** Knudsen number; free path length; kinetic theory

## 1. Introduction

The problem discussed in this article is not new and has about a century of history. It must be said that for many physical problems devoted to the study of the properties of fine media (fogs, vapor, smoke, dust, etc.), in the vast majority of cases, it is characteristic that their solution is mainly purely empirical or experimental in nature. Although the number of theoretical studies in this direction has been growing exponentially in recent years, the derivation of all the basic equations is based on purely experimental dependencies.

In this paper, we will somewhat depart from the established stereotype of solving problems in this direction and use the general principles of the theory of non-equilibrium processes using a dissipative function $\dot{Q} = T\dot{S}$ as the main description parameter, where $T-$ is the equilibrium temperature, $S-$ is the entropy and the "dot" above the letter traditionally indicates differentiation in time.

At the same time, we consider the system conservative and its full volume $V$ is permanent. It is the sum of the volumes of the water drop $V_1$ and the surrounding volume of gas $V_2$. That is $V = V_1 + V_2 = const$. The value $Q$ as defined $\Delta Q = T\Delta S$ is the total amount of heat transferred to the system according to the first law of thermodynamics when $\Delta \dot{Q} = \dot{Q} = 0$. In the non-equilibrium case that we are considering, $\dot{Q} = T\dot{S} \neq 0$.

Before proceeding to the analytical part of the problem, it should be noted that the main purpose of this study is to demonstrate a mathematical tool based on the use of the method of conserving the full power of the system in question, the general descriptive approach of which was described in detail in study [1]. In study [2], this approach was applied already in the framework of solving a specific problem. A comparison with the traditional results described in detail in the monograph [3] showed the full validity of this approach. Furthermore, it is worth emphasizing that all problems solved in this direction (see for example studies [4–9]) can be described using the proposed method. This indicates the relevance of our study which is devoted to the study of the evaporation of a liquid droplet from an alternative point of view.

## 2. Derivation of the Basic Equation and It Analysis

To get the equation we need, we will use the law of conservation of power which is elementarily obtained from the law of conservation of energy. Indeed, according to the law of conservation of energy $\sum_i E_i = const$. Differentiating this equality over time we have

$\frac{d}{dt}\sum_i E_i = \sum_i W_i = 0$, where $\dot{E}_i = W_i$.

We will define this system in the form of two phase components, one of which is a gas and the other is a liquid droplet. Taking into account the interaction of gas phase molecules and molecules in a droplet at the boundary of their contact, the balance equation can be represented in the only way as the sum of three terms (it is easy to understand that there are simply no other components in the problem being solved):

$$T\frac{d}{dt}\int_{V_1} s_1 dV + T\frac{d}{dt}\int_{V-V_1} s_2 dV + \frac{d}{dt}\int_{\sigma} \alpha d\sigma = 0 \tag{1}$$

where $s_1-$ is the droplet entropy attributed to the unit of its volume, $s_2-$ is the entropy of the unit of volume of the gas phase surrounding the liquid droplet including the molecules of the already evaporated substance of the droplet, $V_1-$ is the variable volume of the droplet, $V = V_1 + V_2 = const-$ is the total volume occupied by the droplet and gas, $\sigma-$ is the surface area of the droplet, $T-$ is the temperature, $\alpha-$ is the surface tension coefficient. The Equation (1) describes the total power balance of the conservative system under study: drop + gas.

Looking ahead a little, we note that as a result of solving Equation (1), we will come to the correct answer both qualitatively and quantitatively which is not in contradiction with the well-known result given in the traditional Fuchs monograph [3] but complements it with the qualitatively new result obtained below. This also answers the question of the relevance of our approach when comparing it with the results of other authors working in this direction (see the articles mentioned above [4–9]).

Once again, we emphasize that the principle of preserving the total full power of any dissipative system (and it does not matter whether it is closed or open) leads to correct equations which was strictly proved in concrete physical examples in study [1]. We will not reproduce the details of this study here since in this message we are talking about a specific isolated case which will now be considered in detail.

Performing a simple differentiation in (2) we find
(All variables appearing in Equation (2) are explained after Equation (1)).

$$T\dot{S}_1 + Ts_1\dot{V}_1 + T\dot{S}_2 - Ts_2\dot{V}_1 + \alpha\dot{\sigma}_1 = 0 \tag{2}$$

Introducing here the latent heat of vaporization.
we get from (2)

$$\Delta Q_V = T(s_1 - s_2). \tag{3}$$

$$T\dot{S}_1 + T\dot{S}_2 + \Delta Q_V \dot{V}_1 + \alpha\dot{\sigma}_1 = 0 \tag{4}$$

Our problem now is to calculate the first two terms included in Equation (4). According to the definition of entropy in the language of the distribution function which according to the evidence given in ref. [10] is considered valid for both liquids and gases, we have

$$S_1 = -\frac{1}{Z_1}\int n_1 \ln\frac{n_1}{e} d^3 p. \tag{5}$$

where $n_1-$ is the non-equilibrium function of the distribution of liquid molecules in the momenta, $Z_1 = \int \overline{n}_1 d^3 p$ is the normalization factor and $\overline{n}_1 = e^{-\frac{\varepsilon_1(p)-\mu_1(P,T)}{T}}$ is the equilibrium distribution function, where $P-$ is a pressure.

The ratio (5) according to its definition in [10] is fair in both equilibrium and non-equilibrium cases. Note also that for convenience and reduction of the record, we will consider the Boltzmann constant to be equal to one. That is, we believe that $k_B = 1$. This can always be done since in the final result, in which the temperature will appear in order to obtain the correct dimension, it will only need to be multiplied by $k_B$.

The relationship between the equilibrium and non-equilibrium distribution functions is determined from the conditions of continuity allowing it to be written that $\lim\limits_{t \to \infty} n(t) = \overline{n}$.

$\varepsilon_1(p) = \frac{p^2}{2m_1}$ is the kinetic energy of molecules in a liquid, $\mu_1(P, T)-$ is their chemical potential.

Quite similarly we have

$$S_2 = -\frac{1}{Z_2} \int n_2 \ln \frac{n_2}{e} d^3 p. \tag{6}$$

Considering the temperature constant (that is $Z_1 = const$, $Z_2 = const$), the differentiation of Formulas (5) and (6) in time leads us to the following relations

$$\dot{S}_1 = -\frac{1}{Z_1} \int \dot{n}_1 \ln n_1 d^3 p, \dot{S}_2 = -\frac{1}{Z_2} \int \dot{n}_2 \ln n_2 d^3 p. \tag{7}$$

According to the Boltzmann kinetic equation we have the right to write that

$$\dot{n}_1 = L_1(n_1, n_2), \dot{n}_2 = L_2(n_2, n_1). \tag{8}$$

where $L_1(n_1, n_2), L_2(n_2, n_1)-$ are the integrals of collisions of liquid and gas molecules at the boundary of their contact, respectively.

As for the "internal" collision integrals, that is collisions of gas molecules with each other and liquid molecules with each other in accordance with the Bogolyubov hierarchical principle which allows moving to quasi-equilibrium distribution functions (see (5) and (6)) and "uncoupling" the corresponding correlators, they can be ignored. This remarkable principle makes it possible to find solutions to a variety of problems, the number of which is currently extremely large.

In formal mathematical language, this means that the relaxation times for internal intermolecular collisions $\tau_{11}, \tau_{22}$ are much smaller than the collision times of gas and liquid molecules $\tau_{12}, \tau_{21}$ at the external interface of contact.

That is $\tau_{11} \ll \tau_{12}, \tau_{22} \ll \tau_{21}$. Only when this condition is met do we have the right to introduce a quasi-equilibrium (equilibrium) distribution function into consideration.

Taking into account (6)–(8), Equation (4) will take the following form

$$-\frac{T}{Z_1} \int L_1(n_1, n_2) \ln n_1 d^3 p - \frac{T}{Z_2} \int L_2(n_2, n_1) \ln n_2 d^3 p + \Delta Q_V \dot{V}_1 + \alpha \dot{\sigma}_1 = 0 \tag{9}$$

We will look for the solution of kinetic equations using the BGK method [11] according to which the collision integrals can be replaced with approximate expressions

$$L_1 \approx -\frac{\delta n_1}{\tau_{12}}, L_2 \approx -\frac{\delta n_2}{\tau_{21}}. \tag{10}$$

where $\tau_{12}-$ is the relaxation time of liquid molecules when they are scattered on gas molecules and $\tau_{21}-$ is the relaxation time of gas molecules when they are scattered on liquid molecules. It is quite clear that these times are different. We will now find corrections to the distribution function $\delta n_{1,2}$ due to interaction.

According to kinetic Equation (8), we have

$$\dot{n}_1 = \frac{\partial n_1}{\partial t} + \mathbf{v} \cdot \nabla n_1 + \mathbf{F} \cdot \frac{\partial n_1}{\partial \mathbf{p}} = -\frac{n_1 - \overline{n}_1}{\tau_{12}}. \tag{11}$$

Since we are looking for a stationary solution, then $\frac{\partial n_1}{\partial t} = \frac{\partial n_2}{\partial t} = 0$. In addition, it should be considered that the force $\mathbf{F} = 0$.
Finally,

$$\mathbf{v} \cdot \nabla n_1 = -\frac{n_1 - \overline{n}_1}{\tau_{12}}. \tag{12}$$

Similarly,

$$\mathbf{v} \cdot \nabla n_2 = -\frac{n_2 - \overline{n}_2}{\tau_{21}}. \tag{13}$$

We will search for solutions of Equations (12) and (13) by the method of successive approximations assuming that

$$n_1 = \overline{n}_1 + \delta n_1, \, n_2 = \overline{n}_2 + \delta n_2. \tag{14}$$

Therefore, we get

$$\begin{aligned} \mathbf{l}_{12} \cdot \nabla \delta n_1 + \delta n_1 &= -\mathbf{l}_{12} \cdot \nabla \overline{n}_1, \\ \mathbf{l}_{21} \cdot \nabla \delta n_2 + \delta n_2 &= -\mathbf{l}_{21} \cdot \nabla \overline{n}_2. \end{aligned} \tag{15}$$

where vectors of free path lengths $\mathbf{l}_{12} = \mathbf{v}\tau_{12}$, $\mathbf{l}_{21} = \mathbf{v}\tau_{21}$ are introduced.

It is convenient to search for the solution of Equation (15) by decomposing the desired functions into the Fourier integral. Indeed, we have for an arbitrary (so far) function

$$f(\mathbf{r}) = \int\limits_{-\infty}^{\infty} e^{i\mathbf{k}\mathbf{r}} f_{\mathbf{k}} \frac{d^3\mathbf{k}}{(2\pi)^3}. \tag{16}$$

where by the symbol of a one-dimensional integral we mean a three-dimensional integral, $f_{\mathbf{k}}-$ Fourier image of function $f$. Substituting (16) into any of the Equation (15) we easily find $\int (1 + i\mathbf{k}\mathbf{l}) \delta n_{\mathbf{k}} \frac{d^3\mathbf{k}}{(2\pi)^3} = -\mathbf{l} \cdot \nabla \int \overline{n}_{\mathbf{k}} e^{i\mathbf{k}\mathbf{r}} \frac{d^3\mathbf{k}}{(2\pi)^3}$.

From where

$$\delta n_{\mathbf{k}} = -i\frac{(\mathbf{k} \cdot \mathbf{l})\overline{n}_{\mathbf{k}}}{1 + i\mathbf{k} \cdot \mathbf{l}}. \tag{17}$$

where $\overline{n}_{\mathbf{k}}-$ Fourier is the image of the equilibrium distribution function of molecules $\overline{n}(\mathbf{r})$.

Substituting solution (17) into definition (16), now we obtain the correction to the equilibrium distribution function that interests us

$$\delta n = -\frac{i}{(2\pi)^3} \int \frac{(\mathbf{k} \cdot \mathbf{l})\overline{n}_{\mathbf{k}}}{1 + i\mathbf{k} \cdot \mathbf{l}} e^{i\mathbf{k}\mathbf{r}} d^3\mathbf{k}. \tag{18}$$

Here and further we simplify the recording of the Fourier integral by omitting the limits of integration. To calculate the resulting integral, it is convenient to use the following artificial technique. Let us represent function $\frac{1}{1+i\mathbf{k}\mathbf{l}}$ as an integral

$$\frac{1}{1 + i\mathbf{k}\mathbf{l}} = \int\limits_{0}^{\infty} e^{-x(1+i\mathbf{k}\mathbf{l})} dx. \tag{19}$$

Then from (18) it follows

$$\delta n = -\frac{i}{(2\pi)^3} \int\limits_{0}^{\infty} e^{-x} dx \int (\mathbf{k} \cdot \mathbf{l}) \overline{n}_{\mathbf{k}} e^{i\mathbf{k}(\mathbf{r}-x\mathbf{l})} d^3\mathbf{k}. \tag{20}$$

Further since

$$\overline{n}_{\mathbf{k}} = \int \overline{n}\left(\mathbf{r}'\right) e^{-i\mathbf{k}\mathbf{r}'} d^3\mathbf{r}'. \tag{21}$$

Then after substituting the Fourier image (21) into the solution (20), we will have as a result a simple rearrangement of the multipliers

$$\delta n = -\frac{i}{(2\pi)^3}\int\limits_0^\infty e^{-x}dx\int \overline{n}\left(\mathbf{r}'\right)d^3\mathbf{r}'\int (\mathbf{k}\cdot\mathbf{1})e^{i\mathbf{k}(\mathbf{r}-\mathbf{r}'-x\mathbf{1})}d^3\mathbf{k}. \tag{22}$$

To calculate the internal integral appearing here we proceed as follows. Let us write it down as $\int e^{i\mathbf{k}(\mathbf{R}-\mathbf{1}x)}(\mathbf{k}\mathbf{l})d^3\mathbf{k} = i\frac{\partial}{\partial x}\int e^{i\mathbf{k}(\mathbf{R}-\mathbf{1}x)}d^3\mathbf{k} = i(2\pi)^3\frac{\partial}{\partial x}\delta(\mathbf{R}-\mathbf{1}x).$
where $\delta(x)-$ is the delta function and $\mathbf{R} = \mathbf{r}-\mathbf{r}'$.

As a result, it follows from (22) $\delta n = \int\limits_0^\infty e^{-x}dx\frac{\partial}{\partial x}\int \overline{n}\left(\mathbf{r}'\right)\delta\left(\mathbf{r}-\mathbf{r}'-\mathbf{1}x\right)d^3\mathbf{r}' = \int\limits_0^\infty e^{-x}\frac{\partial}{\partial x}\overline{n}$
$(\mathbf{r}-\mathbf{1}x)dx.$

$$\delta n = \int\limits_0^\infty e^{-x}\frac{\partial}{\partial x}\overline{n}(\mathbf{r}-\mathbf{1}x)dx = \int\limits_0^\infty e^{-x}\overline{n}(\mathbf{r}-\mathbf{1}x)dx - \overline{n}(\mathbf{r}). \tag{23}$$

We take the resulting integral using integration in parts. Really
Remembering now the translational transfer operator, namely the rule $\overline{n}(\mathbf{r}-x\mathbf{1}) = e^{-x\mathbf{1}\cdot\nabla}\overline{n}(\mathbf{r}).$

$$\delta n = \int\limits_0^\infty e^{-x(1+\mathbf{1}\cdot\nabla)}\overline{n}(\mathbf{r})dx - \overline{n}(\mathbf{r}). \tag{24}$$

we get from (23)
Therefore, for the desired corrections in our case, we obtain such solutions of Equation (15)

$$\begin{cases} \delta n_1 = \int\limits_0^\infty e^{-x(1+\mathbf{1}_{12}\cdot\nabla)}\overline{n}_1(\mathbf{r})dx - \overline{n}_1(\mathbf{r}), \\ \delta n_2 = \int\limits_0^\infty e^{-x(1+\mathbf{1}_{21}\cdot\nabla)}\overline{n}_2(\mathbf{r})dx - \overline{n}_2(\mathbf{r}). \end{cases} \tag{25}$$

and, therefore, in accordance with (9) and (10), we find

$$\frac{T}{Z_1}\int\frac{\delta n_1}{\tau_{12}}\ln\overline{n}_1 d^3p + \frac{T}{Z_2}\int\frac{\delta n_2}{\tau_{21}}\ln\overline{n}_2 d^3p + \Delta Q_V\dot{V}_1 + \alpha\dot{\sigma}_1 = 0 \tag{26}$$

where corrections $\delta n_1$, $\delta n_2$ are given by solutions (25).

By virtue of the definition of the equilibrium distribution functions, then the dissipative balance equation follows from (26)

$$-\frac{T}{Z_1}\int\frac{(\varepsilon_1-\mu_1)}{\tau_{12}}\delta n_1 d^3p - \frac{T}{Z_2}\int\frac{(\varepsilon_2-\mu_2)}{\tau_{21}}\delta n_2 d^3p + \Delta Q_V\dot{V}_1 + \alpha\dot{\sigma}_1 = 0 \tag{27}$$

Note that the last term in (27) is conveniently represented as $\int\alpha dS = \overline{\varepsilon}_1 N_1 = \overline{\varepsilon}_1\int c_1 dV_1.$ where $\overline{\varepsilon}_1-$ is some average energy per one particle of a liquid, $c_1-$ is their concentration. In accordance with (25), the solution can be written as an infinite series
$\delta n = \int\limits_0^\infty e^{-x(1+\mathbf{1}\cdot\nabla)}\overline{n}(\mathbf{r})dx - \overline{n} =$

$$= \int\limits_0^\infty e^{-x}\left\{1 - x\mathbf{1}\cdot\nabla + \frac{x^2}{2}(\mathbf{1}\cdot\nabla)^2 - \frac{x^3}{3!}(\mathbf{1}\cdot\nabla)^3 + \dots\right\}\overline{n}(\mathbf{r})dx - \overline{n}.$$

$$\begin{aligned} \delta n &= \left[1 - \mathbf{1}\cdot\nabla + (\mathbf{1}\cdot\nabla)^2 - (\mathbf{1}\cdot\nabla)^3 + \dots\right]\overline{n} - \overline{n} = \\ &= \left[-\mathbf{1}\cdot\nabla + (\mathbf{1}\cdot\nabla)^2 - (\mathbf{1}\cdot\nabla)^3 + (\mathbf{1}\cdot\nabla)^4\dots\right]\overline{n}. \end{aligned} \tag{28}$$

Integrating each term here by $x$, we come to this solution

Where, for the sake of brevity of the record solutions (25) are presented using the uniform notation $\delta n$ and $\mathbf{l}$, that is $\delta n = \{\delta n_1, \delta n_2\}$ and $\mathbf{l} = \{\mathbf{l}_{12}, \mathbf{l}_{21}\}$. If we now substitute solution (28) into the balance Equation (27), then due to momentum integration

$$
\begin{aligned}
-\frac{T}{Z_1} \int \frac{(\varepsilon_1 - \mu_1)}{\tau_{12}} \left[ (\mathbf{l}_{12} \cdot \nabla)^2 + (\mathbf{l}_{12} \cdot \nabla)^4 + (\mathbf{l}_{12} \cdot \nabla)^6 \dots \right] \overline{n} d^3 p - \\
-\frac{T}{Z_2} \int \frac{(\varepsilon_2 - \mu_2)}{\tau_{21}} \left[ (\mathbf{l}_{21} \cdot \nabla)^2 + (\mathbf{l}_{21} \cdot \nabla)^4 + (\mathbf{l}_{21} \cdot \nabla)^6 \dots \right] \overline{n} d^3 p + \Delta Q_V \dot{V}_1 + \alpha \dot{\sigma}_1 = 0
\end{aligned}
\tag{29}
$$

all odd degrees $(\mathbf{l} \cdot \nabla)$ will disappear and instead of (27) we get

Leaving in (29) only the terms quadratic in the free path length and taking into account the explicit form of the equilibrium distribution function, as a result of elementary differentiation we come to the following equation

$$
\left. \begin{aligned}
-\frac{(T - \overline{\varepsilon}_1 - \mu_1)}{T} \frac{l_{12}^2}{\tau_{12}} \left( \Delta \mu_1 + \frac{(\nabla \mu_1)^2}{T} \right) - \\
-\frac{(T - \mu_2)}{T} \frac{l_{21}^2}{\tau_{21}} \left( \Delta \mu_2 + \frac{(\nabla \mu_2)^2}{T} \right) + \Delta Q_V \dot{V}_1 + \alpha \dot{S}
\end{aligned} \right|_{r=R} = 0
\tag{30}
$$

Since the entropy continuity condition must be fulfilled at the boundary of the two phases in the absence of chemical reactions, it is quite clear that the following equality takes place

$$
\Delta Q_V = T(s_1 - s_2)|_{r=R} = 0
\tag{31}
$$

As you can see, this condition is true if the temperature is constant. At the same time, it is quite clear that the equality of entropies at the contact boundary of a droplet and a gas mixture does not at all mean equality of their specific heat capacities since from a formal point of view, equality (31) should be written in a slightly different form namely as

$$
s_1|_{r=R-0} = s_2|_{r=R+0}.
\tag{32}
$$

where the limits are taken to the left and right of the contact boundary.

Therefore, due to the piecewise smoothness of entropy, an additional condition for temperature derivatives of entropy follows from (32) which also binds the heat capacities of both phases. This means that the following equality must take place

$$
c_1|_{r=R-0} = c_2|_{r=R+0} + \Delta c.
\tag{33}
$$

where $\Delta c$ represents the final jump in the heat capacity at the interface of both phases and the isobaric heat capacity is introduced here in accordance with the generally accepted definition [10] $c_i = T \left( \frac{\partial s_i}{\partial T} \right)_P$ where the index $i = 1, 2$ numbers the phases.

As for the physical side of Equation (30), it is immediately necessary to emphasize that as soon as we introduce the concept of variable entropy, we automatically proceed to take into account the dissipative properties of matter. That is, in the non-equilibrium case which is described by Equation (30), the entropy increase condition takes place (the famous $H-$ Boltzmann theorem [10]). As it will become clear now, taking into account the interaction between the molecules of both phases that is the transfer of energy from water molecules to gas molecules and vice versa leads to the destruction of the weak surface tension of a droplet. For an analytical description of this process, it is necessary to focus on the remarkable property of any natural physical phenomenon such as the hierarchy of relaxation times [11–13].

Indeed, in order of magnitude, the free path length of molecules in a liquid $l_{12}$ turns out to be significantly less than the free path length of gas molecules $l_{21}$, that is the inequality $l_{12} \ll l_{21}$ holds.

This means that in terms of the hierarchy of times by virtue of the condition $\tau_{12} \ll \tau_{21}$ which actually follows from the condition $\overline{n}_1 >> \overline{n}_2$ where $\overline{n}_1, \overline{n}_2 -$ are the average concen-

trations of liquid and gas molecules, respectively, the main evaporation process belongs to the first term in (30) and it is this important fact that allows us to neglect the second term.

Otherwise, the first process as the fastest one has already occurred and the droplet has begun to evaporate and the second one has not yet had time to begin. This, however, does not mean at all that it does not contribute to the evaporation process; at a later period of time, this contribution may manifest itself but only if the droplet has not had time to evaporate by this point in time.

Thus, taking into account the condition of continuity of entropy at the contact boundary according to (32) and taking into account all that has been said from Equation (30), we come to such an equation

$$-\frac{(T - \bar{\varepsilon}_1 - \mu_1)}{T}\frac{l_{12}^2}{\tau_{12}}\left(\Delta\mu_1 + \frac{(\nabla\mu_1)^2}{T}\right) + \alpha\dot{S}\Bigg|_{r=R} = 0 \tag{34}$$

Note also that for the chemical potentials of both phases at the boundary of their contact, the following equilibrium condition must also be met

$$\mu_1|_{r=R} = \mu_2|_{r=R}. \tag{35}$$

Since $\dot{S} = 8\pi R\dot{R}$, we find the following from (34)

$$8\pi\alpha R\dot{R} = \frac{(\mu_1 - \bar{\varepsilon}_1 - T)}{T}\frac{l_{12}^2}{\tau_{12}}\left(\Delta\mu_1 + \frac{1}{T}\left(\frac{\partial\mu_1}{\partial R}\right)^2\right). \tag{36}$$

Due to the fact that the distribution of the inhomogeneous chemical potential at the contact of two media (see [14]) obeys the equation

$$\Delta\mu + \frac{\mu}{\delta^2} - \frac{\xi\mu^3}{\delta^2 T^2} = 0 \tag{37}$$

where $\delta-$ is the length of the inhomogeneity satisfying inequality $\delta \ll l_{\min}$ where $l_{\min} = \min\{l_{12}, l_{21}\}$ and $\xi-$ is a certain coefficient leading to a correct solution (see Formula (38)), then in the one-dimensional case, we obtain the following from Equation (37)

$$\mu(r) = \frac{\mu_1 + \mu_2}{2} - \frac{\mu_1 - \mu_2}{2}th\left(\frac{\delta r}{\delta}\right). \tag{38}$$

Therefore, at the contact boundary we have

$$\frac{\partial\mu}{\partial r}\Bigg|_{r=R} = \frac{\mu_2 - \mu_1}{2\delta}. \tag{39}$$

and thus Equation (39) takes the following form

$$8\pi\alpha R\dot{R} = -\frac{(\mu_1 + \bar{\varepsilon}_1 - T)}{T}\frac{l_{12}^2\mu_1}{\tau_{12}\delta^2}\left(\frac{\mu_1}{4T}\left(1 - \frac{\mu_2}{\mu_1}\right)^2 + \frac{\xi\mu_1^2}{T^2} - 1\right). \tag{40}$$

from where after direct integration taking into account the initial condition $R(0) = R_0$ we get

$$R = \sqrt{R_0^2 - D_T t}. \tag{41}$$

where the diffusion coefficient is

$$D_T = \frac{(\mu_1 + \bar{\varepsilon}_1 - T)}{4\pi\alpha T}\frac{\mu_1 l_{12}^2}{\tau_{12}\delta^2}\left(\frac{\mu_1}{4T}\left(1 - \frac{\mu_2}{\mu_1}\right)^2 + \frac{\xi\mu_1^2}{T^2} - 1\right). \tag{42}$$

Equality (41) means that the evaporation time of the liquid droplet we are interested in must be from the condition of equality to zero of the root expression that is

$$t_{vap} = \frac{R_0^2}{D_T}. \tag{43}$$

A characteristic change in the size of the evaporating droplet (41) is shown in Figure 1.

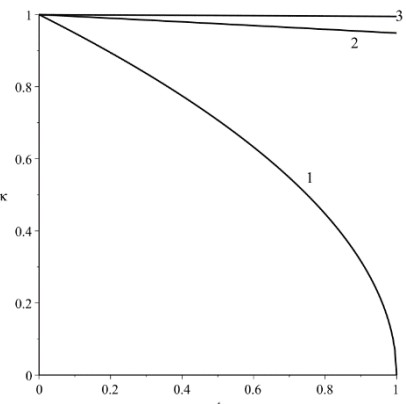

**Figure 1.** The dependence of the radius liquid droplet from time. The curve 1—$\mu = 1$, curve 2—$\mu = 0.1$, curve 3—$\mu = 0.01$, where $\kappa = \frac{R}{R_0}$, $\mu = \frac{D_T}{R_0^2}$.

As for the relaxation time $\tau_{12}$, it can be easily estimated based on the following formula (see [14])

$$\frac{1}{\tau_{12}} = \frac{2r_2^2 \overline{n}_1}{3\pi\sqrt{2\pi}} \frac{m_1^2 \overline{\mu}_1}{(m_1 + m_2)^3} \sqrt{\frac{m_2}{T}}. \tag{44}$$

where $r_2-$ is the radius of a gas molecule, $\overline{\mu}_1-$ is their average chemical potential, $m_1-$ is the mass of a water molecule, $m_2-$ is the mass of a gas molecule, $\overline{n}_1-$ is the average concentration of water molecules. In order of magnitude, it follows from (44) that $\tau_{12} \approx 10^{-10}$ s. A similar formula holds for relaxation time $\tau_{21}$. It is obtained from formula (44) by formally replacing the indices "1" with "2". It can be shown that in order of magnitude $\tau_{21} \approx 10^{-8}$ s.

The calculation of the evaporation time by formula (42) also dictates the need to substitute the chemical potentials of gas and liquid into it. If we proceed from the general definition of the average energy of a large statistical system of particles, namely $\Omega = \mu(P, T)N$, where $N-$ is the number of particles in the system then for its differential we have

$$d\Omega = \left(\frac{\partial \mu}{\partial T}\right)_P NdT + \left(\frac{\partial \mu}{\partial P}\right)_T NdP + \mu dN. \tag{45}$$

According to example [10] in variables $(T, P, N)$, the Helmholtz energy differential is

$$d\Phi = -SdT + VdP + \mu dN. \tag{46}$$

From the comparison of (45) and (46), we see that

$$S = -N\left(\frac{\partial \mu}{\partial T}\right)_P, V = N\left(\frac{\partial \mu}{\partial P}\right)_T. \tag{47}$$

It is known from [10] that the entropy per particle can be calculated as

$$s = \frac{S}{N} = -\frac{1}{Z}\int n \ln \frac{n}{e} d^3\mathbf{p}. \tag{48}$$

where $Z = \int \overline{n} d^3\mathbf{p}$ is the normalization factor and

$$\overline{n} = e^{-\frac{\varepsilon(p)-\mu}{T}} \tag{49}$$

is the equilibrium Maxwell distribution function, $\mathbf{p}-$ is the molecule momentum. Neglecting in (48) the processes of scattering of molecules, we have for entropy

$$S = -\frac{N}{Z}\int \overline{n} \ln \overline{n} d^3\mathbf{p} = \frac{N}{ZT}\int (\varepsilon - \mu)e^{-\frac{(\varepsilon-\mu)}{T}}d^3\mathbf{p}. \tag{50}$$

The chemical potentials in exponential exponents under the integral in (53) and in the normalization factor will decrease and as a result of a simple calculation, we will come to this answer

$$S = N\left(\frac{3}{2} - \frac{\mu}{T}\right). \tag{51}$$

Remembering now definition (47), we obtain the following differential equation for determining $\mu$

$$\left(\frac{\partial \mu}{\partial T}\right)_P = \frac{\mu}{T} - \frac{3}{2}. \tag{52}$$

Simple integration leads us to the following result

$$\mu(P, T) = C(P)T - \frac{3}{2}T \ln T. \tag{53}$$

where dependence $C(P)$ is easily found from the second relation in (47), that is

$$V = N\left(\frac{\partial \mu}{\partial P}\right)_T = NT\frac{dC}{dP}. \tag{54}$$

Since the Clapeyron–Mendeleev equation $PV = NT$ holds for an ideal gas, we immediately get from here that

$$C(P) = A + \ln P. \tag{55}$$

where $A-$ is the constant.

Assuming $A = 1$ and substituting (55) into (53), we find the desired dependence

$$\mu(P, T) = T + T \ln\left(\frac{P}{P_0}\right) - \frac{3}{2}T \ln\left(\frac{T}{T_0}\right). \tag{56}$$

where $T_0$, $P_0-$ is the temperature and pressure under normal conditions that is $T_0 = 300$ K, $P_0 = 1 atm = 10^5$ Pa. That is, for the gas phase, the chemical potential is determined using (56) as

$$\mu_2 = T + T \ln\left(\frac{P_2}{P_0}\right) - \frac{3}{2}T \ln\left(\frac{T}{T_0}\right). \tag{57}$$

As for a droplet of water, it is very problematic to use the gas approximation for it and in this case you can use for example the Van der Waals equation. As a result, the chemical potential can also be calculated analytically but we will not do this now but will proceed to the estimation of the evaporation time considering for simplicity that $\mu_1 : \mu_2$. Note, by the way, that this ratio is quite correct.

To numerically estimate the evaporation time, we will use the general expression (40).

The physical parameters present in Formula (40) can be selected as follows: $\alpha = 70\frac{erg}{cm^2}$, $\mu_1 : \mu_2 = 6 \cdot 10^{-14}$ erg, $T = 300$ K $= 4 \cdot 10^{-14}$ erg, $R_0 = 5 \cdot 10^{-1}$ cm, $\tau_{12} = 10^{-10}$ s, $\delta \approx 10^{-6}$ cm.

Note that in all the estimates below we will use the Gaussian system of units.

As a result, we get that

$$t_{vap} = \tau_{12} \frac{4\pi\alpha R_0^2 \delta^2}{\left(\frac{\mu_1}{4T}\left(1-\frac{\mu_2}{\mu_1}\right)^2 + \frac{\xi\mu_1^2}{T^2} - 1\right)\left(\frac{\mu_1+\bar{\varepsilon}_1}{T} - 1\right)\mu_1 l_{12}^2} \approx$$
$$\approx 10^{-10}\frac{4\pi\cdot70\cdot25\cdot10^{-2}\cdot10^{-12}}{3\cdot6\cdot10^{-14}\cdot10^{-10}} = \frac{2\cdot70\cdot25}{3} \approx 1.15\cdot10^3 \text{ s.} \tag{58}$$

That is, a droplet of water with a diameter of five millimeters evaporates in about twenty minutes. Result (60) is in full correlation with the traditional formula given in [3] which gives us reason to assert the correctness of the results obtained above.

Looking at Equation (36), we quite clearly see before us an equation of the type of thermal conductivity equation with a thermal conductivity coefficient $\chi$ or a diffusion-type equation with a diffusion coefficient $D$ which are determined in order of magnitude by the coefficient of the right side of Equation (36), that is

$$D \sim \chi \sim \frac{l_{12}^2}{\tau_{12}} = \frac{v_{1T}^2 \tau_{12}^2}{\tau_{12}} = v_{1T}^2 \tau_{12}. \tag{59}$$

This remarkable result is evidence that the evaporation process is purely dissipative in nature and under isothermal conditions is determined by the heterogeneity of the chemical potential at the interface between liquid and gas. In light of the above, it can be argued that according to (59), the described evaporation effect is nothing more than isothermal diffusion. In fact, the problem of the analytical description of the droplet evaporation process can be considered solved by evaluation (58).

## 3. Comparison of the Obtained Formulas with Traditional Results

The theoretical approach described above should be compared with the approach described for example in the traditional monograph of Fuchs [3]. In this regard, it should be emphasized right away that the mentioned monograph is entirely based on the interpretation of purely empirical dependencies, that is, dependencies obtained experimentally. However, the formulas given in it allow us to draw some parallels with the theoretical analysis given just above.

Indeed, if we enter the Sherwood number by the formula (see [3,14])

$$Sh = \frac{I_f}{2\pi RD(c_0 - c_\infty)}. \tag{60}$$

where $I_f-$ is the evaporation rate having dimension $\frac{g}{s}$, $D-$ is the diffusion coefficient with dimension $\frac{cm^2}{s}$, $c_0-$ is the vapor concentration in the immediate vicinity of the droplet (its dimension is $\frac{g}{cm^3}$), $c_\infty-$ is the vapor concentration at infinity with the same dimension, then in the case of a stationary droplet, the Sherwood number is exactly 2. Using the empirical dependence (60), it is easy to determine the dependence of the radius of the evaporating droplet on time. Assuming that $I_f = \dot{m}$, where $m = \rho_k V = \frac{4\pi}{3}\rho_k R^3$ is the droplet mass and taking into account that $Sh = 2$, we have the following from (60) $4\pi\rho_k R^2\dot{R} = 4\pi RD(c_0 - c_\infty)$.

or

$$R\dot{R} = \frac{D(c_0 - c_\infty)}{\rho_k}. \tag{61}$$

from where the solution is immediately obtained in the following form

$$R(t) = \sqrt{R_0^2 - D_{eff}t}. \tag{62}$$

where the effective diffusion coefficient is

$$D_{eff} = \frac{D(c_0 - c_\infty)}{\rho_k}. \tag{63}$$

Comparing (62) with our solution (41), we see their complete identity.

However, our diffusion coefficient (42) and diffusion coefficient (63) obtained using the empirical formula differ quite a lot from each other qualitatively. Although it is clear that in order of magnitude, they both give the correct value of the evaporation time of a stationary droplet provided that in Formula (63) the difference is $c_0 - c_\infty$ chosen equal to $1 \frac{\text{g}}{\text{cm}^3}$ and the diffusion coefficient is set equal as in our theory to value $D = 5 \cdot 10^{-5} \frac{\text{cm}^2}{\text{s}}$.

In principle, this is quite understandable since the rigorous analytical solution of the problem based on the equation of conservation of the sum of dissipative functions (1) and the experimentally obtained dependence (60) are based on two different physical assumptions.

## 4. Dynamics of Droplet Passage through A Hot Medium

In the event that a purely technical problem is set related to extinguishing a fire, for example, a burning transformer box, we need to provide an analytical solution to this problem and describe the dynamics of droplets passing through the flame to the surface of the boiling transformer oil taking into account all the basic physical conditions.

If we assume, for example, that the velocity of water from the hose is equal to $3 \cdot 10^3 \frac{\text{cm}}{\text{s}}$, and the distance that the water jet passes to the source of ignition is put equal say $10^3$ cm, then the time of passage of the jet $\Delta t_b$ will be about three-tenths of a second.

Based on estimate (61), it can be assumed that the complete evaporation of a droplet with a diameter of half a centimeter occurs in about an hour; therefore, the droplet does not actually have time to evaporate and passing through the flame, it hits the surface of the oil with almost the same size. It is possible for water droplets to reach the oil surface if the obvious inequality is met

$$\Delta t_B = \frac{h}{v_W} \le t_{vap} = \frac{R_0^2}{D_T} \tag{64}$$

Whence it follows that the droplet size must obey the condition

$$R_0 \ge \sqrt{D_T \frac{h}{v_W}}. \tag{65}$$

As can be seen from the above assessment, the situation is not quite simple in terms of analytical determination of the most effective droplet size. In fact, if we achieve the droplet size, for example, such $R_0 = \sqrt{D_T \frac{h}{v_W}}$, then it simply evaporates quickly without having time to reach the oil surface. This means that here the problem of determining the optimal size of the droplet arises, which despite its small size will still have time to reach the oil surface evaporating directly on it which is the main criterion in the conditions of ignition of oil transformers.

This means that the following strict inequalities must be met

$$\overline{R} < R_0 < R. \tag{66}$$

where $\overline{R} = \sqrt{D_T \frac{h}{v_W}}$ and the value of the right side of inequality (66) $R$ must be found. To calculate it, we will use the equation of motion of a droplet in the gravity field taking into account the drag force and the buoyant force in the following form (see, for example, study [15–17])

$$\dot{u} + \frac{u}{\tau(R)} = g \left( 1 - \frac{\rho_2}{\rho_1} \right). \tag{67}$$

where the attenuation coefficient due to the consideration of the Stokes resistance force is

$$\frac{1}{\tau(R)} = \frac{9\eta_2}{2\rho_1 R^2(t)}. \tag{68}$$

Recall that index "1" refers to a droplet and index "2" refers to a gas. Function $R(t)$ is given by dependency (41) which is convenient to write taking into account (43) in the following form

$$R(t) = \sqrt{R_0^2 - D_T t}. \tag{69}$$

Taking into account (68), Equation (67) can be presented in a more convenient form as

$$\dot{u} + \frac{u}{\tau_0 \left(1 - \frac{t}{t_{vap}}\right)} = g\left(1 - \frac{\rho_2}{\rho_1}\right). \tag{70}$$

where $\frac{1}{\tau_0} = \frac{9\eta_2}{2\rho_1 R_0^2}$ and $t_{vap} = \frac{R_0^2}{D_T}$. The solution of Equation (70) is trivial. In fact, by solving a homogeneous equation we get $\frac{du}{u} = \frac{dt}{\tau_0 \left(\frac{t}{t_{vap}} - 1\right)}$.

hence

$$u = C\left(\left|\frac{t}{t_{vap}} - 1\right|\right)^{\beta}. \tag{71}$$

where $\beta = \frac{t_{vap}}{\tau_0}$ is the index. Considering constant $C$ as a function of time and then substituting (71) into Equation (70), we find $\dot{C} = \frac{g}{\left(\left|\frac{t}{t_{vap}} - 1\right|\right)^{\beta}}\left(1 - \frac{\rho_2}{\rho_1}\right)$.

that is

$$C(t) = C - \frac{g t_{vap}}{(\beta - 1)\left(\left|\frac{t}{t_{vap}} - 1\right|\right)^{\beta - 1}}\left(1 - \frac{\rho_2}{\rho_1}\right). \tag{72}$$

where $C-$ is the integration constant. Substituting (72) into (71), we find the following solution

$$u = C\left(\left|\frac{t}{t_{vap}} - 1\right|\right)^{\beta} + \frac{g t_{vap}\left|\frac{t}{t_{vap}} - 1\right|}{(1 - \beta)}\left(1 - \frac{\rho_2}{\rho_1}\right). \tag{73}$$

To determine constant $C$, we use initial condition $u(0) = u_0$. As a result, $C = u_0 - \frac{g}{(1-\beta)}\left(1 - \frac{\rho_2}{\rho_1}\right)$ and the final solution will take the following form

$$u = u_0\left(1 - \frac{t}{t_{vap}}\right)^{\beta} + \frac{g\left(t_{vap} - t\right)}{(1 - \beta)}\left(1 - \frac{\rho_2}{\rho_1}\right)\left[1 - \left(1 - \frac{t}{t_{vap}}\right)^{\beta - 1}\right]. \tag{74a}$$

In a dimensionless view, we have

$$y = (1 - x)^{\beta} + \frac{\lambda(1 - x)}{(1 - \beta)}(1 - \varepsilon)\left[1 - (1 - x)^{\beta - 1}\right]. \tag{74b}$$

where $x = \frac{t}{t_{vap}}, \lambda = \frac{g t_{vap}}{u_0}, y = \frac{u}{u_0}, \varepsilon = \frac{\rho_2}{\rho_1}$ and $0 \leq x \leq 1$.

The dependence (74b) is shown in Figure 2.

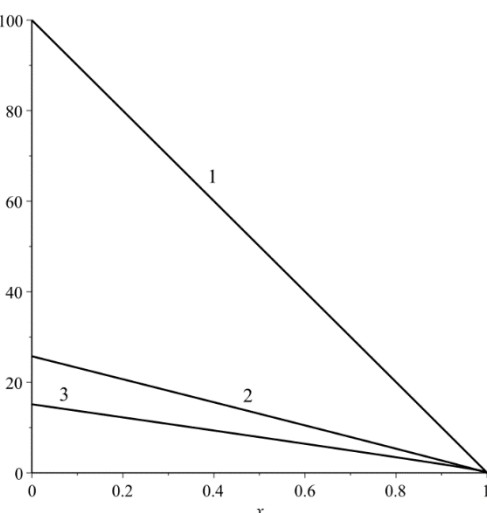

**Figure 2.** The dependence of droplet velocity from time. The curve 1—$\beta = 0.9$, curve 2—$\beta = 0.6$, curve 3—$\beta = 0.3$.

It is taken into account here that always $t < t_{vap}$. Integrating (74) in time, we find the droplet path length

$$z(t) = C_1 + \frac{u_0 t_{vap}}{\beta + 1}\left(1 - \frac{t}{t_{vap}}\right)^{\beta+1} + \frac{g t_{vap}^2 \left(1 - \frac{t}{t_{vap}}\right)^2}{2(1 - \beta)}\left(1 - \frac{\rho_2}{\rho_1}\right)\left[1 - \frac{2}{1 + \beta}\left(1 - \frac{t}{t_{vap}}\right)^{\beta-1}\right]. \tag{75}$$

From the initial condition $z(0) = h$, we get

$$C_1 = h - \frac{u_0 t_{vap}}{\beta + 1} + \frac{g t_{vap}^2}{2}\left(1 - \frac{\rho_2}{\rho_1}\right). \tag{76}$$

therefore,

$$
\begin{aligned}
z(t) = {} & h + \frac{u_0 t_{vap}}{\beta+1}\left[\left(1 - \frac{t}{t_{vap}}\right)^{\beta+1} - 1\right] + \\
& + \frac{g t_{vap}^2}{2(1-\beta)}\left(1 - \frac{\rho_2}{\rho_1}\right)\left\{\left(1 - \frac{t}{t_{vap}}\right)^2\left[1 - \frac{2}{1+\beta}\left(1 - \frac{t}{t_{vap}}\right)^{\beta-1}\right] + 1 - \beta\right\}.
\end{aligned} \tag{77a}
$$

In a dimensionless view, the dependence (77a) can be written as

$$
\begin{aligned}
Z = {} & 1 + \frac{\gamma}{\beta+1}\left[(1 - x)^{\beta+1} - 1\right] + \\
& + \frac{\sigma}{2(1-\beta)}(1 - \varepsilon)\left\{(1 - x)^2\left[1 - \frac{2}{1+\beta}(1 - x)^{\beta-1}\right] + 1 - \beta\right\}
\end{aligned} \tag{77b}
$$

where the parameters are

$\varepsilon = \frac{\rho_2}{\rho_1}, \gamma = \frac{u_0 t_{vap}}{h}, \sigma = \frac{g t_{vap}^2}{h}, Z = \frac{z(t)}{h}$,

$\gamma = 10, \sigma = 10, \varepsilon = 0.01$

The dependence (77b) is illustrated in Figure 3.

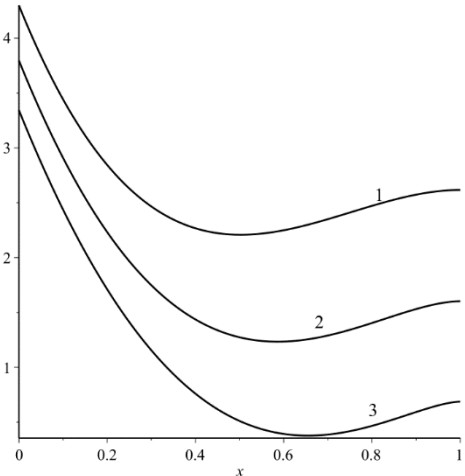

**Figure 3.** The dependence of dimensionless length of flying from time. The curve 1—$\beta = 2$, curve 2—$\beta = 1.3$, curve 3—$\beta = 0.9$.

From condition $z = 0$, we can calculate the time of the droplet movement to the oil surface of our interest taking into account its evaporation. This algebraic equation must be solved under the condition that $t < t_{vap}$. Therefore, by decomposing function $z(t)$

$$z(t) \approx h - u_0 t + \frac{g t_{vap} \cdot t}{2(1 + \beta)} \left(1 - \frac{\rho_2}{\rho_1}\right). \tag{78}$$

by degrees of ratio $\frac{t}{t_{vap}}$, we find approximately

From here taking into account evaporation, the travel time will be

$$\Delta t_f = \frac{h}{u_0 - \frac{g t_{vap}}{2(1+\beta)} \left(1 - \frac{\rho_2}{\rho_1}\right)}. \tag{79}$$

During this period of time, the droplet size decreases and becomes equal according to (69)

$$R\left(\Delta t_f\right) = \sqrt{R_0^2 - \frac{D_T h}{u_0 - \frac{g t_{vap}}{2(1+\beta)} \left(1 - \frac{\rho_2}{\rho_1}\right)}}. \tag{80}$$

Thus, from solution (80), taking into account the explicit form for $\beta$, it follows that the initial droplet size should be

$$R_0 \geq \sqrt{\frac{D_T h}{u_0 - \frac{g t_{vap} \tau_0}{2(\tau_0 + t_{vap})} \left(1 - \frac{\rho_2}{\rho_1}\right)}}. \tag{81}$$

In turn, remembering that $\tau_0 = \frac{2\rho_1 R_0^2}{9\eta_2}$, we get a biquadrate inequality to determine the possible values $R_0$. In fact, from (84) we get

$$R_0^2 \geq \frac{D_T h}{u_0 \left[1 - \frac{g \rho_1 R_0^2}{9\eta_2 u_0 \left(\frac{2\rho_1 R_0^2}{9\eta_2 t_{vap}} + 1\right)} \left(1 - \frac{\rho_2}{\rho_1}\right)\right]}. \tag{82}$$

It is convenient to bring this inequality to the following form

$$R_0^2 \geq \frac{R_3^2}{\left[ 1 - \frac{R_0^2}{R_2^2 \left( 1 + \frac{R_0^2}{R_1^2} \right)} \right]}. \tag{83}$$

where $R_1 = 3\sqrt{\frac{\eta_2 t_{vap}}{2\rho_1}}$, $R_2 = 3\sqrt{\frac{\eta_2 u_0}{g(\rho_1 - \rho_2)}}$, $R_3 = \sqrt{\frac{D_T h}{u_0}}$.

Moreover, the hierarchy of these parameters is as follows

$$R_3 \ll R_2 \ll R_1. \tag{84}$$

Therefore, using condition $R_0 \ll R_1$ and leaving only one in the lowest fraction, we easily solve the simplified biquadrate inequality which leads us to the following condition

$$R_3 \leq R_0 \leq R_2 \ll R_1. \tag{85}$$

Substituting explicit expressions for radii from (83), we find

$$\sqrt{\frac{D_T h}{u_0}} \leq R_0 \leq 3\sqrt{\frac{\eta_2 u_0}{g(\rho_1 - \rho_2)}} \ll 3\sqrt{\frac{\eta_2 t_{vap}}{2\rho_1}}. \tag{86}$$

From where we find the condition for the initial velocity of the droplet

$$u_0 \geq u_{cr}. \tag{87}$$

where $u_{cr} = \frac{1}{3}\sqrt{\frac{D_T g h(\rho_1 - \rho_2)}{\eta_2}}$.

From the example, we can take the following parameters $D_T = 5 \cdot 10^{-5} \frac{\text{cm}^2}{\text{s}}$, $g = 10^3 \frac{\text{cm}}{\text{s}^2}$, $h = 10^3$ cm, $\eta_2 = 10^{-2} \frac{\text{g}}{\text{cm} \cdot \text{s}}$, $\rho_1 = 1 \frac{\text{g}}{\text{cm}^3}$, $\rho_2 = 0.01 \frac{\text{g}}{\text{cm}^3}$.

then we get

$$u_{cr} = 1,3 \cdot 10^3 \frac{\text{cm}}{\text{s}}. \tag{88}$$

The numerical value (88) is in full accordance with the known practical results; therefore, the solution of the problem can be considered complete in accordance with estimates (86) and (87).

## 5. Conclusions

Summing up the above research, it is worth noting the following three important points:

We have proposed a theory of evaporation of droplets of a finely dispersed medium based on the condition of preserving the system power (the dissipated energy cannot disappear without a trace, but transforms into something);

We have proposed an analytical description of the complex dynamics of the droplet motion in a high-temperature medium taking into account its evaporation;

We have given numerical estimates of the droplet size and initial jet velocity corresponding to experimental values using the obtained formulas.

**Funding:** This research received no external funding.

**Institutional Review Board Statement:** Not applicable.

**Informed Consent Statement:** Not applicable.

**Acknowledgments:** The author thanks S.B. Bogdanova for help in Figures.

**Conflicts of Interest:** The authors declar no conflict of interest.

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
