# Peer review of "On Some Theoretical Aspects of The Evaporation Process of a Droplet and Its Optimal Size When Extinguishing Fires"

_inventions, doi:10.3390/inventions8010035_

Round 1

Reviewer 1 Report

The authors of manuscript “On Some Theoretical Aspects of The Evaporation Process of a Droplet and its Optimal Size When Extinguishing Fires” proposed a mathematical model of droplet evaporation using kinetic approach. This model is based on the conservation of the full power of the vapor-liquid system.

From the formulation it is not clear if the author considers evaporation of pure liquid in its own vapor or in the gas phase another gas (condensable or not is present). Many statements in the text are given without any justification or appropriate citation.

I cannot recommend the manuscript for publication in journal.

The referee has the following questions and suggestions:

1.     On page 1 Q is not defined.

2.     On page 2 the first sentence needs to be reformulated

3.     Under what conditions Eq. (5) could be applied to describe a liquid?

4.     How equilibrium and non-equilibrium function n are related.

5.     Why you can assume the Boltzmann constant to be equal to 1

6.     Capital P letter on page 2 is a pressure?

7.     Boltzmann kinetic equation is derived to describe the gas flow behaviors and not the liquid flow because of the binary collusion hypothesis

8.     Equation (8) needs to be justified: why the “internal” collisions can be neglected.

Author Response

Notes to Reviewer

The author is grateful to the reviewer for the comments made to my article. Following the ordinal numbering of these remarks I would like to answer them as follows. 

  1. The value as defined is the total amount of heat transferred to the system according to the first law of thermodynamics  when . In the non-equilibrium case we are considering .
  2. The first sentence on page 2 is reformulated as follows.

To get the equation we need, we will use the law of conservation of power, which is elementarily obtained from the law of conservation of energy. Indeed, according to the law of conservation of energy . Differentiating this equality over time we have , where . 

  1. According to the evidence given in [10] formula (5) is considered to be valid for both liquids and gases. The corresponding comment is also given in the corrected text of the article.
  2. The relationship between equilibrium and non-equilibrium distribution functions due to the fulfillment of the continuity condition is as follows .
  3. Regarding Boltzmann's constant.

For convenience and reduction of the record we can consider the Boltzmann constant equal to one that is put that . This can always be done (in many works it is often used) since in the final result in which the temperature will appear to obtain the correct dimension it will only need to be multiplied by .

  1. it's pressure. The relevant comment is given in the text.
  2. We use an approach that is described in detail in the classic monograph [15]. The authors describe the possibility of applying the Boltzmann equation when taking into account binary collisions directly to liquids.
  3. The ability to "neglect" internal collisions is dictated by the following considerations based on the principle of the hierarchy of relaxation times.

In formal mathematical parlance, this means that the relaxation times  for internal intermolecular collisions are significantly less than the collision times  between gas and liquid molecules at the outer contact boundary in a very narrow boundary layer with some conditional width . That is . Only when this condition is met, we have the right to introduce a quasi-equilibrium (equilibrium) distribution function into consideration. 

 All corrections made are highlighted in purple.

Some syntax and grammatical errors have also been corrected in the text of the work. And in a number of places and some phrases. 

    Sincerely, Author

Reviewer 2 Report

Include a short sketch at the beginning to show the volumes involved in the integration of eq. (1). Precisely, declare which type of balance is imposed in that equation.

In eq. 58 replace the cyrilic symbols for minute. Check the same in the rest of the equations.

I would finally suggest to include some comprehensive plots showing the trends of the evaporation times and/or most relevant physical variables with respect to the main function variables (size, temperature difference, surface tension or similar).

Author Response

Notes to Reviewer 

The author is grateful to the reviewer for the comments made to his article on the essence of which I would like to note the following.

  1. At the beginning of the work is placed a complete list of volumes that are involved in solving the problem. It is also explained what kind of balance was applied.
  2. In the ratio (58) and later in the text of the article all temporal dimensions are replaced by seconds.
  3. Added three graphic illustrations that illustrate temporal dependencies (41), (74), (77)

The corresponding corrections in the corrected version of the work are made in red. 

Sincerely,

Author
